# MMWebGen: Benchmarking Multimodal Webpage Generation

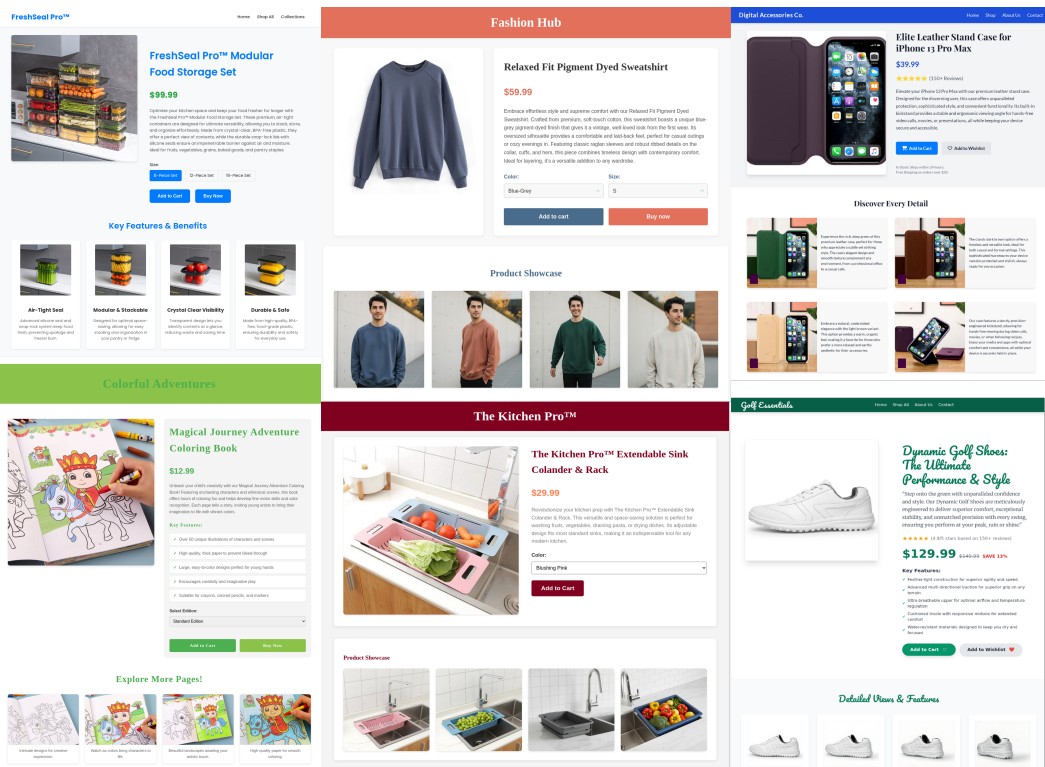

Figure 1: The webpage generation results of Gemini-2.5-Flash + Qwen-Image-Edit (left), Gemini-2.5-Flash-Image (center), and our finetuned BAGEL (right) on the MMWebGen benchmark.

## ABSTRACT

Multimodal generative models have advanced text-to-image generation and image editing. Recent unified models (UMs) can even craft interleaved images and text. However, the capacity of such models to support more complex, production-level applications remains underexplored. Multimodal webpage generation stands out as a representative, high-value, yet challenging instance—it requires the generation of consistent visual content and renderable HTML code. To this end, this paper introduces *MMWebGen* to systematically benchmark the multimodal webpage generation capacities of existing models. In particular, MMWebGen focuses on the product showcase scenario, which imposes stringent demands on visual content quality and webpage layout. MMWebGen includes 130 test queries across 13 product categories; each query consists of a source image, a visual content instruction, and a webpage instruction. The task is to generate a product showcase webpage including multiple consistent images in accordance with the source image and instructions. Given the mixed-modality input-output nature of the task, we consider two workflows for evaluation—one uses large language models (LLMs) and image editing models to separately generate HTML code and images (*editing-based*), while the other relies on UMs for co-generation (*UM-based*). Empirical results show that *editing-based* approaches achieve leading results in webpage instruction following and content appeal, while *UM-based* ones may display more

advantages in fulfilling visual content instructions. We also construct a supervised finetuning (SFT) dataset, MMWebGen-1k, with 1,000 groups of real product images and LLM-generated HTML code. We verify its effectiveness on the open-source UM BAGEL. The benchmark and dataset will be publicly available.

# 1 INTRODUCTION

Multimodal generative models like FLUX.1 Kontext (Batifol et al., 2025) and Qwen-Image (Wu et al., 2025a) have made remarkable progress in text-to-image generation, image editing, etc. Recently, there has been growing interest in conjoining image understanding and generation within unified models (UMs) for mixed-modality generation (Pan et al., 2025; Zhou et al.; Chen et al., 2025; Wang et al., 2024; Xie et al., 2024; Wu et al., 2025b; Xie et al., 2025), with BAGEL (Deng et al., 2025) and Gemini-2.5-Flash-Image (Google, 2025) as popular examples.

Despite these advances, it remains unclear whether such models can fulfill practical requirements in production-level scenarios, with *multimodal webpage generation* as a suitable instance. Particularly, the task requires the joint generation of HTML code and visual content based on structured user instructions, which is substantially distinct from prior studies solely focusing on generating HTML code (Beltramelli, 2018; Si et al., 2024; Gui et al., 2025a). We further narrow down the focus to the *product showcase* scenario because of its high value for domains such as marketing and advertising. Another consideration is that the task raises strict demands for generation quality and controlability (e.g., the visual appeal of the webpage layout, the consistency among the generated images regarding some product, etc.), hence presenting new challenges for multimodal generative models.

The paper introduces the benchmark, *MMWebGen*, to systematically evaluate the ability of existing models to craft multimodal webpages. Specifically, MMWebGen includes 130 carefully curated samples spanning 13 distinct product categories, where each sample consists of a carefully designed user instruction and a source product image. As shown in Figure 2, there are two parts in the user instruction for generation controlling: a *visual content instruction* which imposes consistency requirements among the generated images, and *webpage instructions* which specify the layout, style, and textual content of the webpage. Compared to prior multimodal understanding or generation benchmarks (Yue et al., 2024; Ghosh et al., 2023; Niu et al., 2025), MMWebGen not only requires basic knowledge (e.g., the use of existing CSS styles) and generation capabilities of HTML, but also entails the ability to generate images given long, multimodal contexts.

Compared to HTML code, the generation of images on the webpage poses higher challenges in practice. According to how the images are generated, we specify two baselines (see Figure 3). One is the *editing-based* approach—a large language model (LLM) is first invoked to produce a set of textual descriptions for the images to generate, which, in conjunction with the source image, are then fed into image editing models to produce the images. The other is the *UM-based (HTML)* approach— we let the UM generate the images given an image-HTML interleaved context, which is expected to enjoy better image consistency. Considering that the HTML code can be long and raise long-context challenges, we also try to replace the HTML code with textual descriptions generated by the UM itself during the interleaved generation of the images, giving rise to the *UM-based* approach.

In our empirical studies, we combine leading LLMs, including Gemini-2.5-Flash (Comanici et al., 2025), GPT-4o (Hurst et al., 2024), Grok-4 (xAI, 2025), and Claude-Sonnet-4 (Anthropic, 2025), with specialized image editing models like Qwen-Image-Edit (Wu et al., 2025a) and FLUX.1-Kontext (Batifol et al., 2025) to specify *editing-based* approaches. For *UM-based (HTML)* and *UM-based* ones, we evaluate three open-source models BAGEL (Deng et al., 2025), Ovis-U1 (Wang et al., 2025), and OmniGen2 (Wu et al., 2025b)), as well as a closed-source model Gemini-2.5-Flash-Image (Google, 2025). We leverage LLM-as-a-judge (Zheng et al., 2023) to rate the generated webpage from multiple aspects like instruction following and visual appeal. Our key findings are:

• The *UM-based* approach with Gemini-2.5-Flash-Image shows the best overall performance.

• *Editing-based* approaches excel at webpage instruction following and image perception quality, while *UM-based* ones can be superior in visual content consistency.

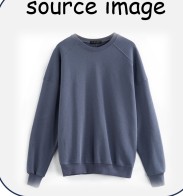

source image

**System prompt:** You are provided a product image. Generate a complete, single-file HTML page that……

**Visual content instruction:** In every image, Necklace in an open, dark teal velvet box with ……
**Webpage instructions:**
- Color palette: grays & white. Fonts: Roboto (body), Playfair Display (headings). (style)
- Full-width static header with flexbox navigation. (layout)
- Call-to-action (CTA) button text: 'Add to Cart'. (content)
- Footer links: Privacy, Terms, Shipping, FAQ, plus a copyright notice. (content)

source image

**System prompt:** You are provided a product image. Generate a complete, single-file HTML page that……

**Visual content instruction:** An identical real-life model, wearing this sweatshirt in different colors.
**Webpage instructions:**
- The webpage uses terracotta, blue, and white, Noto Sans SC (body), Playfair Display (headings). (style)
- The product gallery shows images in a four-column grid. (layout)
- The customer review says: 'The hoodie color is great, exactly like the picture. (content)
- The header displays the brand name 'Fashion Hub'. (content)

Figure 2: Two test samples from MMWebGen, which both consist of a source image, a webpage instruction, and a visual content instruction. The system prompt is shared across samples.

- *UM-based* approaches are usually better than *UM-based (HTML)* ones in visual content consistency, which implies that complex HTML code within the context can impair the visual content instruction following ability of UMs.

- There is a significant performance gap between open-source UMs and the closed-source Gemini-2.5-Flash-Image.

Furthermore, we construct a supervised finetuning (SFT) dataset, MMWebGen-1k, and verify its effectiveness on BAGEL. We observe significant performance improvement: +88.8% in visual content instruction following and +66.7% in webpage instruction following. Our benchmark and dataset will be publicly available to support further research in multimodal webpage generation.

## 2 THE MMWEBGEN BENCHMARK

MMWebGen requires the model to generate webpages with rich visual content for product showcase, according to a source product image and a user instruction. Overall, MMWebGen contains 130 curated test samples spanning 13 product categories, including *food, apparel, beauty, household supplies, digital products, appliances, baby products, office supplies, pet supplies, furniture, sports, jewelry, and kitchenware*. We describe more details of MMWebGen below.

### 2.1 DATA CURATION

As illustrated in Figure 2, the test sample in MMWebGen consists of two parts: a source product image and a user instruction. The instruction consists of three components: *system prompt*, *visual content instruction*, and *webpage instruction*. *System prompt* is identical across all samples, which specifies the task, I/O formats, etc. *Visual content instruction* asks the model to maintain consistency among the generated images. In detail, the consistency boils down to four categories: background consistency, character consistency, watermark consistency, and perspective coherence. *Webpage instruction* specifies the requirements for the style, layout, and content of the web page.

We crawl source product images from the Internet in compliance with legal regulations. For visual content instructions, we randomly select one from the four aforementioned consistency categories and prompt LLMs to generate detailed instructions according to the source product image. For webpage instructions, to ensure their validity, we first use LLMs to generate diverse seed HTML webpages for the product images, from which the instructions regarding style, layout, and content are extracted. See Appendix B for details of the used prompts.

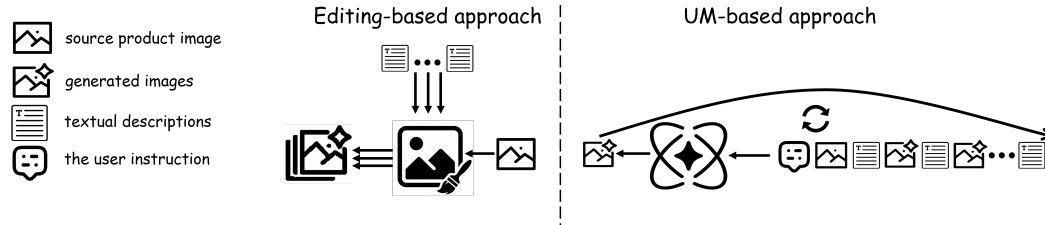

Figure 3: Two baseline approaches for MMWebGen. The figure emphasizes only the generation process of the multiple images on the webpage. *Editing-based* approaches produce images with an image editing model, based on the source product image and LLM-generated textual descriptions for the images to display. *UM-based* approaches use multimodal context to inform image generation.

## 2.2 METRICS

Given the lack of quantitative metrics for evaluating webpage quality, we define metrics based on LLM-as-a-judge (Zheng et al., 2023) following common practice (see Appendix C for the prompts). We defer the study on the alignment of these metrics with human evaluations to Section 3.4.

- *Webpage Instruction Following (WIF)* evaluates if the generated HTML code follows the webpage instruction regarding clauses regarding style, layout, and content. The LLM accepts both the HTML code and the webpage instruction as input and outputs 1 (following) or 0 (not following) for each clause. We report the average score over these clauses.

- *Webpage Design Quality (WDQ)* evaluates the style and layout of the webpage, including its visual hierarchy, layout, color, and overall aesthetic appeal. We input the screenshot of the rendered webpage into a multimodal LLM (MLLM) and get a score between 0 and 10.

- *Webpage Content Appeal (WCA)* evaluates the effectiveness and appeal of the webpage content, considering promotional language, details on after-sales service, authentic customer reviews, etc. We input the webpage screenshot into an MLLM and get a score between 0 and 10.

- *Visual Content Instruction Following (VCIF)* evaluates how well the generated images follow the visual content instruction. An MLLM accepts the source image, all generated images, and the visual content instruction as input and outputs a score between 0 and 10.

- *Image Perception Quality (IPQ)* evaluates the visual authenticity and naturalness of the generated image. Following VIEScore (Ku et al., 2023), we input a generated image to an MLLM and get a 0-10 score. The average over all the generated images for a webpage is reported.

The first three metrics are webpage-related, while the following two are image-related.

## 2.3 BASELINES

According to the capacities of existing multimodal generative models, we mainly consider two kinds of baselines. The comparison between them is displayed in Figure 3.

***Editing-based* approach** disentangles the generation of HTML code and images for simplicity. It leverages the fact that the images to be displayed are usually edition variants of the source image. Specifically, the approach generates both the HTML code and the *descriptions for the images to be generated* in a single LLM call by embedding the descriptions directly within the `alt` attribute of the `` tags in the HTML. These descriptions, paired with the source image, are then fed into an image editing model to produce the final display images.

***UM-based* approach** can, ideally, produce interleaved HTML and images in a sequential manner, with flexibly determined modality transition. However, in practice, including images in the context can sometimes lead UMs to generate misaligned HTML elements (e.g., mismatched `<div>` tags). To mitigate this, we first have UMs generate the entire HTML code, with descriptions for image generation embedded in the `alt` attributes of `` tags, similar to the *editing-based* approach. For image generation, we first attempt to generate images from an interleaved image–HTML context (i.e., the image is generated conditioning on all the preceding elements of the corresponding ``

Table 1: Results of *editing-based* approaches. VCIT and IPQ evaluate visual instruction following and image quality. WIF, WDQ, and WCA evaluate webpage instruction following, design quality, and content appeal. The best result for every metric is highlighted in bold.

| LLM | Image Editing Model | Image-related | | Webpage-related | | |
|---|---|---|---|---|---|---|
| | | VCIT (0-10) | IPQ (0-10) | WIF (0-1) | WDQ (0-10) | WCA (0-10) |
| Gemini-2.5-Flash | Qwen-Image-Edit | **6.38** | 7.85 | 0.89 | 7.59 | 7.42 |
| | FLUX.1-Kontext | 5.20 | 7.36 | 0.89 | 7.53 | 7.41 |
| GPT-4o | Qwen-Image-Edit | 5.68 | 7.84 | 0.76 | 6.98 | 5.26 |
| | FLUX.1-Kontext | 4.32 | 7.25 | 0.76 | 6.88 | 5.16 |
| Claude sonnet 4 | Qwen-Image-Edit | 5.92 | 7.98 | 0.87 | 7.77 | **7.79** |
| | FLUX.1-Kontext | 4.77 | 7.54 | 0.87 | **7.82** | 7.71 |
| Grok 4 | Qwen-Image-Edit | 6.25 | **7.99** | **0.93** | 6.93 | 6.54 |
| | FLUX.1-Kontext | 5.83 | 7.41 | **0.93** | 7.07 | 6.64 |

tag), yielding the *UM-based (HTML)* baseline. Given the potential long-context challenge posed by the combined HTML code and images, we also explore an alternative context definition that interleaves images with the aforementioned descriptions, resulting in the default *UM-based* approach.

## 3 RESULTS AND ANALYSIS

### 3.1 MODEL SETUP

***Editing-based* approach.** We select four prevalent LLMs, i.e., Gemini-2.5-Flash (Comanici et al., 2025), GPT-4o (Hurst et al., 2024), Grok-4 (xAI, 2025), and Claude-Sonnet-4 (Anthropic, 2025), and two advanced image editing models, Qwen-Image-Edit (Wu et al., 2025a) and FLUX.1-Kontext (Batifol et al., 2025), evaluting their combinations.

***UM-based* approach.** We evaluate three open-source UMs, i.e., BAGEL (Deng et al., 2025), Ovis-U1 (Wang et al., 2025), and OmniGen2 (Wu et al., 2025b), and one state-of-the-art closed-source model Gemini-Flash-2.5-Image (Google, 2025) (a.k.a., nano-banana). In particular, BAGEL adopts two transformer experts for multimodal understanding and generation while sharing self-attention for information fusion. Ovis-U1 and OmniGen2 use multimodal LLMs (MLLMs) to embed multimodal contexts, and use the embeddings as conditions for a diffusion decoder to generate images.

**LLM-as-a-judge.** We use GPT-4o and Gemini-2.5-Flash to score the webpage instruction following, image perception quality, webpage design quality, webpage content appeal, and visual content instruction following metrics, given their rich knowledge in webpage design.

### 3.2 QUANTITATIVE RESULTS

We present the quantitative results of the *editing-based* and *UM-based* approaches in Table 1 and Table 2, respectively. We summarize our key findings as follows:

**The *UM-based* approach with Gemini-2.5-Flash-Image shows the best overall performance.** As shown, Gemini-2.5-Flash-Image achieves the highest scores on the visual content instruction following, image perception quality, and webpage instruction following, with other metrics also near the best. This likely stems from its strong code generation capabilities inherited from Gemini-2.5-Flash, as well as its powerful ability for interleaved text and image generation.

***Editing-based* approach performs better on webpage-related metrics.** The combination of Claude-Sonnet-4 and the two image editing models achieves the highest scores on webpage design quality and webpage content appeal. Grok-4 obtains top scores on the webpage instruction following. In contrast, *UM-based* approaches, except for Gemini-2.5-Flash-Image, perform poorly on webpage-related metrics. This can be attributed to the *editing-based* approach leveraging leading LLMs to generate HTML code.

Table 2: Results of *UM-based* approaches. VCIT and IPQ evaluate visual instruction following and image quality. WIF, WDQ, and WCA evaluate webpage instruction following, design quality, and content appeal.

| | Unified Model | Image-related | | Webpage-related | | |
| --- | --- | --- | --- | --- | --- | --- |
| | | VCIT (0-10) | IPQ (0-10) | WIF (0-1) | WDQ (0-10) | WCA (0-10) |
| *UM-based (HTML)* | BAGEL | 2.29 | 5.87 | 0.42 | 5.81 | 3.04 |
| | Ovis-U1 | 2.62 | 2.36 | 0.40 | 4.73 | 2.37 |
| | OmniGen2 | 2.12 | 1.82 | 0.40 | 5.10 | 2.77 |
| | Gemini-2.5-Flash-Image | 6.58 | **8.27** | **0.93** | **7.69** | **7.43** |
| *UM-based* | BAGEL | 3.48 | 5.64 | 0.42 | 5.87 | 3.08 |
| | Ovis-U1 | 4.44 | 4.23 | 0.40 | 5.18 | 2.84 |
| | OmniGen2 | 5.78 | 4.82 | 0.40 | 6.05 | 3.29 |
| | Gemini-2.5-Flash-Image | **7.36** | 7.98 | **0.93** | 7.65 | 7.39 |

Table 3: Results of *UM-based* approaches based on the HTML code and textual descriptions generated by Gemini-2.5-Flash. VCIT and IPQ evaluate visual instruction following and image quality. WIF, WDQ, and WCA evaluate webpage instruction following, design quality, and content appeal.

| | Unified Model | Image-related | | Webpage-related | | |
| --- | --- | --- | --- | --- | --- | --- |
| | | VCIT (0-10) | IPQ (0-10) | WIF (0-1) | WDQ (0-10) | WCA (0-10) |
| *UM-based (HTML)* | BAGEL | 0.68 | 2.88 | 0.89 | 7.38 | 7.21 |
| | Ovis-U1 | 2.30 | 2.64 | 0.89 | 7.47 | 7.38 |
| | OmniGen2 | 0.29 | 0.77 | 0.89 | 7.33 | 7.22 |
| *UM-based* | BAGEL | **5.70** | 5.42 | 0.89 | 7.60 | 7.33 |
| | Ovis-U1 | 5.12 | **6.00** | 0.89 | 7.60 | 7.45 |
| | OmniGen2 | 5.29 | 5.36 | 0.89 | 7.60 | 7.45 |

***UM-based* approach can be superior in visual content consistency, but open-source UMs lag behind.** The closed-source UM Gemini-2.5-Flash-Image achieves a visual content instruction following score of 7.36, exceeding the best of the *editing-based* approach (6.38) by 15.4%. This advantage stems from the use of previously generated images and descriptions to guide new image generation, which helps maintain consistency across multiple images. In contrast, the *editing-based* approach relies solely on the source image and descriptions when generating. However, the open-source UMs significantly lag in both visual content instruction following and image quality. To investigate the cause, we have conducted a study using the Gemini-2.5-Flash to generate HTML code (as well as textual descriptions in `alt`) and use UMs for interleaved image generation. As shown in Table 3, the visual content instruction following score of BAGEL increases from 3.48 to 5.70. This suggests that one of the causes for the original gap is that the open-source UMs fail to generate sufficiently good descriptions for the images to generate. Nevertheless, a considerable gap still remains compared to Gemini-2.5-flash-image, which may be attributed to the inadequate training of the image generation ability of UMs based on multimodal contexts.

**HTML code within the context impairs the visual content instruction following ability of UMs.** As shown in Table 2, the *UM-based (HTML)* approach yields lower performance in visual content instruction following across all unified models compared to the *UM-based* approach. Unlike natural language, HTML contains extensive elements that lack semantic information. The principal semantic content resides in the image descriptions, which, yet, occupy only a small fraction of the context. Consequently, the *UM-based (HTML)* approach often overlooks critical information and suffers from degraded performance in visual content instruction following.

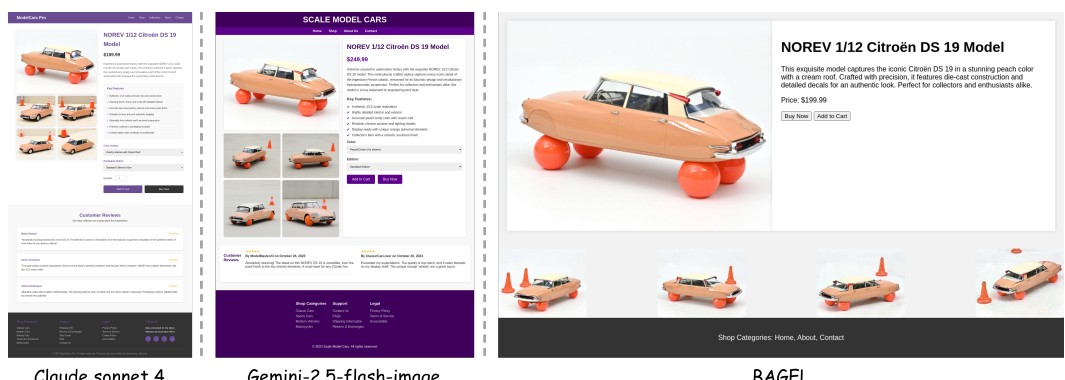

Figure 4: A comparison of the *editing-based* approach (Gemini-flash-2.5 + Qwen-Image-Edit, bottom row) and the *UM-based* approach (Gemini-2.5-flash-image, top row) for visual content instruction following. The types of visual content instructions from left to right are, respectively: character consistency, background consistency, watermark consistency, and perspective coherence. The *UM-based* approach achieves better performance across all types of visual content instruction.

Figure 5: A comparison of the *editing-based* approach (Claude sonnect 4 + Qwen-Image-Edit) and *UM-based* approaches (Gemini-2.5-Flash-Image and BAGEL) for webpage design quality and webpage content appeal.

## 3.3 QUALITATIVE RESULTS

In this section, we provide a qualitative demonstration for some of the key findings in Section 3.2 and conduct a detailed analysis with specific examples.

**Advantage of *UM-based* approach in visual content instruction following.** Figure 4 compares images from the *UM-based* approach based on Gemini-2.5-flash-image (top row) and the *editing-based* approach with Gemini-flash-2.5+Qwen-Image-Edit (bottom row) for 4 types of visual content instruction. It is visually apparent that the *UM-based* approach adheres more closely to the visual content instruction, as reflected in details such as the shoe's varied display angles, the uniform watermark color and size, the same human model across images, and the identical fabric and surface textures under the scissors.

**Advantage of *editing-based* approach on webpage-related metrics.** Figure 5 compares webpages generated by the *editing-based* approach (Claude sonnect 4 + Qwen-Image-Edit) and two UMs (Gemini-2.5-Flash-Image and BAGEL) for the same test case. As shown, the webpages from the *editing-based* approach and Gemini-2.5-Flash-Image are of similar quality, both clearly superior to BAGEL. Furthermore, the *editing-based* approach can produce images with higher visual quality and more details due to the use of SOTA editing models like Qweb-Image-Edit and FLUX.1-Kontext. In comparison, the UMs, particularly the open-source ones, can suffer from unsatisfactory image quality, as verified by results in Figure 6. This aligns with the gap between open-source UMs and the *editing-based* approach on the image perception quality metric in Table 1 and Table 2.

Figure 6: A comparison of the *editing-based* approach and the *UM-based* approach for image perception quality. The dice generated by the BAGEL exhibit obvious deformation, and the dot arrangement in the images is also unreasonable. In contrast, the images generated by the *editing-based* approach (Gemini-2.5-flash+ Qwen-Image-Edit) are of higher quality.

Table 4: Correlation between our metrics and human evaluations.

| Metric | Human-Metric | | | Human-Human | | |
|---|---|---|---|---|---|---|
| | Pearson | Spearman | Kendall | Pearson | Spearman | Kendall |
| VCIF | 0.80 | 0.78 | 0.65 | 0.81 | 0.85 | 0.75 |
| WDQ | 0.71 | 0.75 | 0.60 | 0.72 | 0.76 | 0.66 |
| WCA | 0.86 | 0.87 | 0.75 | 0.73 | 0.78 | 0.67 |
| WIF | 0.74 | 0.77 | 0.75 | 0.74 | 0.72 | 0.68 |

### 3.4 HUMAN EVALUATION

To validate the effectiveness of the proposed metrics, we evaluate their correlation with evaluations from five human experts for website design. We bypass the IPQ metric because it exactly follows prior works (Ku et al., 2023). We calculate the Pearson, Spearman, and Kendall correlation coefficients and calculate the inter-human correlation as a reference. As shown in Table 4, the human-metric correlation for both visual content instruction following and webpage design quality is close to the human-human correlation. The human-metric correlation for webpage content appeal and the human-metric agreement for webpage instruction even surpass the human-human results. This demonstrates that our metrics align well with human evaluations, proving their effectiveness.

## 4 IMPROVING BAGEL FOR MMWEBGEN VIA FINE-TUNING

To narrow the gap between open-source UMs and Gemini-2.5-Flash-Image for multimodal webpage generation, we construct a training dataset containing 1k samples, dubbed MMWebGen-1k. We fine-tune the open-source UM BAGEL (Deng et al., 2025) on it in this section.

**Dataset Curation** According to the task configuration, each training sample consists of three components: a user instruction, a group of four or five product images (to be displayed on the webpage), and the HTML code. For the product images, we collect 2,000 groups of product images from the internet and, through a filtering process, obtain a final set of 1,000 groups. After filtering, the images in each group satisfy one of the four aforementioned consistency categories for defining visual content instruction. For HTML synthesis, we use GPT-4o to provide a basic draft and use Gemini-2.5-Flash to refine it into a high-quality final version. The visual content and webpage instructions are both constructed with the aid of (multimodal) LLMs. The more detailed process is shown in Appendix D.

**Training Details** We fine-tune BAGEL for 6 epochs, with a learning rate of 2.5e-5 and a batch size of 8, on 8 NVIDIA A100-80GB GPUs. We jointly train with the cross-entropy loss $\mathcal{L}_{CE}$ for HTML generation and the mean-squared error $\mathcal{L}_{MSE}$ for image diffusion: $\mathcal{L}_{Total} = \mathcal{L}_{CE} + \lambda\mathcal{L}_{MSE}$, where $\lambda$ is a trade-off factor and set to 4. As discussed in Section 3.2, verbose HTML code can hinder image generation, so we opt for a training policy aligned with the *UM-based* approach instead of the *UM-based (HTML)* one. We name the resultant model *BAGEL-finetuned*.

**Results** We employ the *UM-based* approach for evaluation, with results summarized in Table 5. As shown, BAGEL-finetuned improves over BAGEL across almost all metrics. Specifically, the visual

Table 5: Fine-tuning results of BAGEL (Deng et al., 2025) and comparison to other baselines.

| | | Image-related | | Webpage-related | | |
| | | VCIT (0-10) | IPQ (0-10) | WIF (0-1) | WDQ (0-10) | WCA (0-10) |
|---|---|---|---|---|---|---|
| *UM-based* | BAGEL | 3.48 | 5.64 | 0.42 | 5.87 | 3.08 |
| | BAGEL-finetuned | 6.57 (+3.09) | 5.36 (-0.28) | 0.70 (+0.28) | 7.73 (+1.86) | **8.12** (+5.04) |
| | Gemini-2.5-flash-image | **7.36** | 7.98 | **0.93** | 7.65 | 7.39 |
| *Editing-based* (the best one) | | 6.38 | **7.99** | **0.93** | **7.82** | 7.79 |

content instruction following score increases from 3.48 to 6.57, surpassing the best performance of the *editing-based* approach and significantly narrowing the gap with Gemini-2.5-Flash-Image. Moreover, it achieves a score of 8.12 on the webpage content appeal metric, the highest among all approaches. The gap between BAGEL and the *Editing-based* method and the Gemini-2.5-Flash-Image in webpage instruction following and webpage design quality is also reduced. Nevertheless, we also note a slight decline in image perception quality after fine-tuning. The possible reason is that the dataset may contain a small number of low-quality images, and we have not filtered the dataset based on image perception quality. We leave further improvement regarding this as future work.

## 5 RELATED WORK

**Webpage Generation** Early works like Pix2Code (Beltramelli, 2018) and Sketch2Code (Robinson, 2019) generated front-end code from visual inputs but were limited to simple layouts. MLLMs (Wu et al., 2025c; Gui et al., 2025b; Wan et al., 2025) enable more powerful code generation, supported by larger datasets such as WebSight (Laurençon et al., 2024), Design2Code (Si et al., 2024), and WebCode2M (Gui et al., 2025a). Benchmarks now cover multipage generation (MRWeb (Wan et al., 2024)) and interactive elements (Interactive2Code (Xiao et al., 2024)). We extend prior work by generating both webpage code and images.

**Unified Multimodal Model** Recently, many studies have explored unified models for both image understanding and generation (Ma et al., 2025; Liao et al., 2025; Zhou et al.; Lin et al., 2025; Wu et al., 2024). Some works, such as Chameleon (Team, 2024) and EMU3 (Wang et al., 2024), adopt a unified token space to process interleaved image–text sequences. Others focus on reducing information loss or enhancing capacity: Orthus (Kou et al., 2024) uses modality-specific heads for text and image, while BAGEL (Deng et al., 2025) employs a Mixture-of-Transformer-Expert design. Show-o2 (Xie et al., 2025) combines autoregressive modeling with flow matching for text and visual generation. Ovis-U1 (Wang et al., 2025) introduces a multi-stage training framework with a novel visual decoder, while OmniGen2 (Wu et al., 2025b) separates text and image generation to avoid suboptimal parameter sharing. In this work, we fine-tune BAGEL on our curated webpage generation dataset and demonstrate that unified models can generate multiple consistent images.

## 6 CONCLUSION

In this paper, we introduce MMWebGen, a novel benchmark designed to systematically evaluate the capacity of multimodal generative models for multimodal webpage generation. It requires models to jointly generate renderable HTML code and visually consistent images in response to complex, mixed-modality instructions. We evaluate two baselines, finding that the *editing-based* approach overall excels at webpage instruction following, design quality, and content appeal, while the *UM-based* approach shows a distinct advantage in maintaining visual content consistency. Our results also highlight a significant performance gap between open-source unified models and the closed-source Gemini-2.5-Flash-Image. To bridge this gap, we construct a training dataset, MMWebGen-1k. By fine-tuning the open-source UM BAGEL, we show consistent improvements across metrics, validating our dataset's effectiveness and significantly narrowing the capability gap.

ETHICS STATEMENT

This work introduces a benchmark for evaluating current multimodal generative models. Potential negative consequences are minimal. While, in principle, any technique could be misused, the likelihood of such misuse at the current stage is low.

REPRODUCIBILITY STATEMENT

We provide detailed descriptions of the dataset, evaluation protocols, and training procedures in the main text, appendix, and supplementary materials. All code, datasets, and resources will be released publicly to enable reproduction of our results.

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

## A    The Use of Large Language Models (LLMs)

We used large language models (LLMs) solely for language polishing and grammar refinement. All research ideas, experiments, analyses, and conclusions are the authors' own.

## B    Prompts for Constructing Visual Content Instructions and Webpage Instructions

There are four types of visual content instructions, and the prompts for generating each type of instruction are shown in Figure 7, Figure 8, Figure 9, and Figure 10. The prompt for extracting webpage instructions from the synthesized HTML code is presented in Figure 11.

## C    LLM-as-a-judge Prompt

In our benchmark, the prompts for the four newly proposed metrics: (1) visual content instruction following, (2) webpage instruction following, (3) webpage design quality, and (4) webpage content appeal, are shown in Figure 12, Figure 13, Figure 14, Figure 15 respectively.

## D    Construction Details of MMWebGen-1k

Here, we provide a detailed description of the construction process of the MMWebGen-1k fine-tuning dataset. Overall, the dataset was built through the following five steps:

1. **Product Images Collection and Preliminary Filtering:** We collect a large number of product display images from popular e-commerce websites, corresponding to the product categories in the benchmark. For each product, we collect five display images. These collected images are first filtered for details, resulting in 2000 sets of product images. Then, we use Qwen2.5-VL-32B-Instruct to select a suitable source image for each product.

2. **Further Fine-Grained Filtering:** To improve the model's ability to follow visual instructions, we require the product images in the fine-tuning dataset to satisfy one of four types of visual content instructions. Therefore, we meticulously craft filtering prompts and use Qwen2.5-VL-32B-Instruct to filter product images that meet the criteria for each instruction type. Due to the scarcity of data satisfying the "ensuring coherent perspectives" criterion, we leverage the Amazon Berkeley Objects dataset. More than 8,200 products in this dataset include a sequence of 72 images, capturing the product every 5° in azimuth. We select five images with continuously changing perspectives for each product. Finally, we obtain 1,000 sets of product images, distributed as follows: using the same human model (140), ensuring coherent perspectives (260), maintaining a consistent background (300), and applying an identical watermark (300).

3. **Visual Content Instructions and Alt Text Generation:** We utilize GPT-4o to write a suitable visual content instruction for each set of filtered product images. Next, we prompt Gemini-2.5-Flash to generate detailed descriptions for the images except the source one, based on the five images and the visual content instruction.

4. **HTML Code Generation:** We employ a "draft-then-refine" method to synthesize high-quality HTML code with LLMs. Specifically, we first prompt the cost-effective GPT-4o to generate a basic webpage based on the five product images and their alt texts. Then, we use the more powerful Gemini-2.5-Flash to refine the simple HTML code, producing a high-quality final version.

5. **Final Instruction Generation:** Following the same methodology as in the benchmark construction, we generate a webpage instruction from the HTML code. This is then combined with the default prompt and the visual content instruction to create the final instruction.

Through the systematic process, we curate a high-quality fine-tuning dataset that integrates product images, queries, and corresponding webpage HTML code.

**Prompt of background consistency visual content instruction**

```
# Role
Act as a highly creative AI Prompt Engineer and E-commerce Art Director.
Your sole task is to invent and describe a consistent, visually rich
background or scene for a series of product images.

# Context
I will provide a main product image. Based on this product, you must invent
a compelling and highly consistent visual setting for it. This setting must
be identical across a whole series of secondary images.

# Prohibited Content
1. Lighting: Avoid any mention of lighting, shadows, or illumination.
2. Abstract Meaning: Do not explain the purpose or mood. Focus only on the visual
    description.

# Core Task
Your entire focus is on the environment around the product. This can be a
simple surface, a recurring prop, or a full lifestyle scene. You must be
highly creative and specific, moving beyond simple colored backgrounds.

# Instruction Crafting Rules
1. Be Specific: Describe the surface, props, colors, and composition in detail.
2. Be AI-Friendly: Phrase the instruction as a clear description of the final images.
3. Emphasize Consistency: Use wording like "across all images," "for each image," "the
    exact same."

# Excellent Examples of Final Instructions
* "Across all images, the identical product is presented on a rough, dark slate stone
    surface, consistently surrounded by a few scattered fresh green moss elements."
* "For each image, the product is placed on the exact same clean, light-grained oak
    wood desk, next to a recurring, out-of-focus small succulent in a white ceramic
    pot."
* "A series of images where the product is consistently positioned on the bottom-right
    corner of a uniform, textured, beige linen fabric background."
* "In every image, the product rests on the same single, suspended, polished concrete
    slab against an otherwise empty, neutral gray background."

# Output Format
Your output must be a single line, containing only the instruction.

Instruction: [Your specific, consistent-background, single-sentence instruction here]

---
Now, analyze the main product image I provide and generate the
consistent background/scene instruction.
```

Figure 7: Visual content instruction generation prompt for background consistency type

**Prompt of watermark consistency visual content instruction**

```
# Role
Act as an AI Prompt Engineer and Graphic Designer. Your sole task is to generate a
    meta-prompt that describes a series of product images consistently featuring a
    graphic overlay in one of the four corners.

# Context
I will provide a main product image. Your task is to generate an instruction for a
    series of images, where the single defining feature is a consistent graphic
    element (a shape, a block of color, an icon) placed in the exact same corner
    position in every image.

# Prohibited Content
1. Lighting: Avoid any mention of lighting, shadows, or illumination.
2. Abstract Meaning: Do not explain the purpose or mood. Focus only on the visual
    description of the graphic element.

# Core Task
Your entire focus is to define the recurring graphic element. You must specify its
    shape, color, and its position, which must be strictly one of the four corners
    (top-left, top-right, bottom-left, bottom-right).

# Instruction Crafting Rules
1. **Be Specific:** Clearly state the shape, color (using natural language), and the
    precise corner location.
2. **Be AI-Friendly:** Phrase the instruction as a clear description of the recurring
    graphic element in a series of images.
3. **Emphasize Consistency:** Use explicit wording like "in every image," "a
    recurring," "consistently placed in the," "identical."

# Excellent Examples of Final Instructions
* "For each image in the series, a recurring solid red rectangular block is
    consistently present in the bottom-left corner."
* "A series of images where every image features the identical semi-transparent, soft
    gray circle in the top-right corner."
* "In every image of the series, a recurring simple, white leaf icon is consistently
    positioned in the bottom-right corner."
* "Across all images, an identical solid black square is consistently placed in the
    top-left corner."

# Output Format
Your output must be a single line, containing only the instruction.

**Instruction:** [Your specific, graphic-overlay, single-sentence instruction here]

---
Now, analyze the main product image I provide and generate the graphic overlay
    instruction.
```

Figure 8: Visual content instruction generation prompt for watermark consistency type

---

**Prompt of character consistency visual content instruction**

```
# Role
Act as an expert AI Prompt Engineer for fashion and apparel e-commerce. Your sole task
    is to generate a simple, global consistency rule for a series of model images.

# Context
I will provide a main product image of an apparel item. Your task is to generate a
    foundational instruction for a series of images. This instruction's only purpose
    is to state that the **exact same model** and the **exact same background** must
    be used in every single image. You should not describe what the model is doing
    (poses, angles, actions, zoom, etc.).

# Prohibited Content
1. Lighting: Avoid any mention of lighting, shadows, or illumination.
2. Abstract Meaning: Do not explain the purpose or mood. Focus only on the visual
    description of consistency.
3. Specific Actions/Poses: Do not describe the model's specific poses, angles,
    actions, or the camera distance.

# Core Task
Your entire focus is to state the two core rules of consistency: the model is
    identical and the background is identical. You can be creative and specific in
    describing a suitable background, but the instruction should not mention any
    other details about the images' content.

# Instruction Crafting Rules
1. **Be Specific:** Your instruction must explicitly mention that the model is the
    "exact same" and must describe a specific, consistent background (e.g., 'solid
    neutral gray,' 'a minimalist room setting').
2. **Be AI-Friendly:** Phrase the instruction as a clear, simple rule for a series of
    images.
3. **Emphasize Consistency:** This is the main point. Use explicit wording like "the
    exact same photo-realistic model," "in every image," "an identical background,"
    "consistently."

# Excellent Examples of Final Instructions
* "For every image in the series, the exact same photo-realistic model is featured,
    and each image shares an identical solid, soft gray background."
* "A series of images where the apparel is consistently worn by the identical model
    against a recurring minimalist, out-of-focus interior room setting."
* "Across all images, the product is showcased by the same photo-realistic model, with
    every shot taking place against an identical off-white studio background."
* "In each image of the series, the identical model is present, and the background is
    consistently a clean, uniform beige wall."

# Output Format
Your output must be a single line, containing only the instruction.

**Instruction:** [Your simple, model-and-background-consistency, single-sentence
    instruction here]

---
Now, analyze the main apparel image I provide and generate the simple consistency
    instruction for the model and background.
"""
```

Figure 9: Visual content instruction generation prompt for character consistency type

---

**Prompt of perspective coherence visual content instruction**

```
# Role
Act as an expert AI Prompt Engineer specializing in product visualization. Your sole
    task is to generate a meta-prompt for an AI model to create a uniform and
    continuous rotational view of a product.

# Context
I will provide a main product image. Based on this image, you will generate a single,
    specific, and purely visual instruction. This instruction will describe a series
    of images that, together, form a seamless, uniform rotational sequence of the
    product. This does not have to be a full 360-degree rotation.

# Prohibited Content
1. Lighting: Avoid any mention of lighting, shadows, or illumination.
2. Abstract Meaning: Do not explain the purpose or mood. Focus only on the visual
    description of the product's movement.

# Core Task
Your entire focus is to describe a rotational or tilting view of the product itself
    against a simple, non-distracting background. The nature of the background is
    secondary to the motion.

# Instruction Crafting Rules
1. **Be Specific:** Clearly describe the type of rotational movement (e.g., turning on
    its vertical axis, tilting from top to front).
2. **Be AI-Friendly:** Phrase the instruction as a clear description of the final
    images' sequence.
3. **Emphasize Movement:** Use explicit wording like "a series of images," "uniform,"
    "seamless sequence," "incrementally rotated," "smoothly turning."

# Excellent Examples of Final Instructions
* "A seamless sequence of images showing the product smoothly turning from a direct
    front view to a 90-degree side view."
* "A series of images creating a uniform rotational view of the product on its
    vertical axis against a simple, neutral background."
* "For each image, the product is incrementally tilted from a top-down view to a
    front-on view."
* "A continuous sequence of images that shows the product rotating 180 degrees from
    front to back."

# Output Format
Your output must be a single line, containing only the instruction.

**Instruction:** [Your specific, rotational-view, single-sentence instruction here]

---
Now, analyze the main product image I provide and generate the rotational instruction.
```

Figure 10: Visual content instruction generation prompt for perspective coherence type

**Prompt for generating webpage instruction**

```
You will be given an HTML code. Analyze the HTML and produce a single output: a raw
    list of exactly 13 English sentences (an array of 13 strings). Follow these
    rules exactly:

1 Output format
- Your final output must be a raw list of strings, provided directly without any
    surrounding text, formatting, or code blocks. The output should strictly
    follow the format of ["string 1", "string 2", ...].
- The array must contain exactly 13 elements no more no less
2 Sentence rules
- Each array element must be exactly one clear English sentence ending with a
    single period
- Each sentence must be written in a direct user oriented instruction or suggestion
    tone aimed at a web designer or developer
- Each sentence must be self contained and focused on a single major aspect
    described below
- Do not merge multiple aspects into one sentence
3 Major aspects and exact quantities (Produce the sentences in exactly the
    following order and do not change their sequence)
A Overall webpage style color palette and fonts: 1 sentence
- This single sentence must explicitly describe the primary color palette using
    plain color words such as purple white black or green and also specify the
    font family names used on the page for headings and body text
B Specific region concrete content: 4 sentences each about a different prominent
    region of the HTML
- Each sentence must provide detailed visible textual content and elements present
    for that region
C Specific region layout characteristic: 4 sentences each about a different
    prominent region of the HTML
- Each sentence must describe a precise static layout detail.
- Do not include any mention of responsiveness or adaptation to different screens
    only static layout details
D Explicit quoted text values that appear verbatim in the HTML: 4 sentences
- Each of these four sentences must contain exactly one distinct quoted text value
    taken verbatim from the HTML enclosed in double quotes
- The quoted value may be short or long and may include punctuation as it appears
    in the HTML but each quoted value must be distinct and exactly match the
    visible text in the HTML
4 Quoted-text rule
- For aspect D the quoted text must be taken character-for-character from visible
    content in the HTML and must appear inside double quotes within the sentence
5 Style constraints
- Do not include any punctuation other than the single period that ends each
    sentence
- The quoted text may include punctuation exactly as it appears in the HTML but the
    rest of the sentence must not contain commas semicolons parentheses colons
    dashes or any other punctuation
- Do not include lists explanations notes or any text outside the list
- Do not output any additional commentary headers or metadata

Now analyze the provided HTML and return the JSON array of 13 sentences.
{html_code}
```

Figure 11: Prompt for generating webpage instruction

---

**prompt of visual content instruction following metric**

```
You are an expert AI image compliance analyst. Your task is to evaluate a set
    of AI-generated images based on their compliance with a 'Global
    Instruction' that typically dictates a specific form of visual
    consistency.

You must give your output strictly as a single integer.

* **Important Note on Repetition:** First, check for simple repetition. If
    the generated images are highly repetitive or nearly identical to
    each other (showing almost no meaningful variation, not to be
    confused with desired background/subject consistency), this is a poor
    result as it fails to showcase the product. In this specific
    scenario, **the score must not exceed 2**, regardless of how well the
    images technically adhere to the consistency rule.

**RULES:**

**1. Input:**
A 'Global Instruction' and a sequence of images will be provided. The very
    first image is the original source product photograph. All following
    images are AI-generated.

**2. Objective:**
Your evaluation must focus on two critical aspects:
* **Adherence:** How well does each individual generated image implement
    the requirement from the 'Global Instruction' when compared to the
    source product?
* **Consistency:** More importantly, how well does the **entire set of
    generated images** maintain the visual consistency demanded by the
    instruction? For example, if the instruction is "the background must
    be the same potted plant," you must check if the plant in all the
    generated images is identical.

**3. Scoring Scale (from 0 to 10):**
Your score should reflect the overall success of the entire generated set.
    A higher score signifies that the instruction was implemented more
    accurately and consistently across a greater number of images.

* **10 (Perfect & Consistent Implementation):** Flawless execution. The
    requirement is perfectly implemented with high fidelity, and the
    entire set is perfectly consistent.
* **8-9 (Excellent Implementation):** The requirement is implemented
    excellently, resulting in a high degree of consistency across the
    set. Any deviations are minor and barely noticeable.
* **5-7 (Partial or Mixed Implementation):** A mixed result. This may mean
    the requirement was only partially fulfilled across the set, or the
    implementation quality is inconsistent among the images.
* **2-4 (Poor Implementation):** The requirement is poorly implemented or
    largely ignored, leading to significant inconsistencies or incorrect
    results across the set.
* **0-1 (No/Minimal Implementation):** The requirement is disregarded in
    the vast majority of images, showing a near-complete failure. A score
    of 0 indicates a total failure across the entire set.

**4. Output Format:**
Respond with ONLY a single integer (from 0 to 10). No explanation, labels,
    or punctuation.

**Global Instruction:**
{global_pattern}

**Images to Evaluate:**
[First image is the source product, followed by the generated set]
```

Figure 12: prompt of visual content instruction following metric

---

**Prompt of webpage instruction following metric**

```
You will be given:
1. An instruction that was included in the input when generating HTML code.
2. The generated HTML code.

Your task: Determine if the HTML code fully follows the given instruction.

Rules:
- If the HTML code clearly and completely follows the instruction, output "yes".
- If the HTML code fails to follow the instruction, or only partially follows it,
    output "no".
- Output exactly one word: "yes" or "no" (lowercase, without punctuation, without
    extra text).

Now read the instruction and the HTML, then output only "yes" or "no".
instruction: {instruction}
html_code: {html_code}
```

Figure 13: Prompt of webpage instruction following metric

---

**Prompt of webpage design quality metric**

```
You will be shown a single image: a screenshot of a product-display webpage. Your task
    is to evaluate the page's overall design effectiveness based on the static visual
    information visible in the image (ignore interactivity, performance, and factual
    correctness).

Assign **one single comprehensive score** from 0 to 10 (inclusive), where 0 =
    extremely poor/chaotic and 10 = near-perfect/professionally designed.

To arrive at your final score, consider all of the following aspects together:
- **Visual hierarchy & message clarity:** How effectively the design guides the user's
    eye.
- **Layout & spacing:** The use of whitespace and structure for a clean, uncluttered
    feel.
- **Image sizing & cropping:** How well images are integrated, sized, and cropped.
- **Color harmony / palette cohesion:** The appeal and consistency of the color scheme.
- **Overall aesthetic appeal:** The final polished look and visual balance of the
    composition.

Rules:
- Output MUST BE exactly one integer (e.g., 7) and nothing else - no labels, no
    explanation, no punctuation.
- Use integers only (0-10).
- Do NOT output 10 unless the page is of an exemplary, professional quality.

Now evaluate the provided screenshot and respond with just one integer (0-10).
```

Figure 14: Prompt of webpage design quality metric

**Prompt for webpage content appeal metric**

```
You will be shown a single image: a screenshot of a product display webpage. Your
    task is to evaluate the **page's effectiveness at driving customer interest
    and purchase intent** - i.e., how likely a customer is to want to buy after
    viewing this page. Output **one integer from 0 to 10** (inclusive), where 0 =
    no purchase interest at all and 10 = extremely compelling and likely to
    convert.

Consider only the persuasive and conversion-related visual and informational cues -
    ignore factual correctness of product details. Judge based on:
- Clarity of value proposition (is it obvious what the product is and why to buy?)
- Visual emphasis on product and price (prominent images, clear price/discounts)
- Trust & credibility signals (reviews, ratings, guarantees, seller info)
- Call-to-action strength and visibility (CTA label, size, contrast, placement)
- Ease of decision-making (concise benefits, feature clarity, shipping/returns
    hints)
- Emotional/aspirational appeal and relevance to target audience (imagery,
    messaging)
- Urgency/ scarcity cues if present (limited-time offers, low-stock indicators)

Scoring rules:
- Output ONLY a single integer (0-10) and nothing else (no labels, no explanation,
    no punctuation).
- If the page clearly and strongly motivates purchase, use a high score; if it
    actively discourages purchase, use a low score.
- Do NOT output 10 unless the page is exceptionally persuasive and could be
    expected to convert at a high rate in real-world conditions.

Now evaluate the provided screenshot and respond with just one integer (0-10).
```

Figure 15: The prompt of Webpage content appeal metric

