# OpenReview forum: "MMWebGen: Benchmarking Multimodal Webpage Generation"
_ICLR.cc/2026/Conference — ICLR 2026 Conference Withdrawn Submission_

### Official Review · Reviewer_78iK · 2025-10-30

**Soundness:** 2
**Presentation:** 2
**Contribution:** 2
**Rating:** 2
**Confidence:** 4

**Summary:**

This paper addresses the gap in evaluating multimodal webpage generation by proposing the MMWebGen benchmark, which focuses on product showcase scenarios. It includes 130 test samples across 13 product categories, each consisting of a source image, visual content instructions, and webpage instructions. The work compares two technical workflows—editing-based (separate LLM for HTML and image editing models for visuals) and UM-based (unified multimodal models for co-generation)—and constructs the MMWebGen-1k supervised fine-tuning dataset to improve the open-source UM BAGEL. Overall, the study provides a systematic evaluation framework for multimodal webpage generation, with empirical insights into the strengths of different workflows.

**Strengths:**

1. Fills the gap in multimodal webpage generation for product showcase scenarios. Unlike generic webpage generation benchmarks, MMWebGen specifically targets high-value product showcase scenarios with 130 curated test samples across 13 product categories, using a tailored multimodal input to address real-world user needs.

2. Establishes a comprehensive multimodal alignment evaluation framework. It proposes five targeted metrics (WIF, WDQ, WCA, VCIF, IPQ) covering both webpage (instruction following, design, content) and image (consistency, quality) dimensions, and validates their alignment with human evaluations.

**Weaknesses:**

1. Contemporary webpage generation works like Web2Code (NeurIPS 2024) leverage real existing webpages as foundational data for code generation, enabling alignment with real-world webpage structures, styles, and logic. In contrast, MMWebGen’s test samples and training data only rely on a single source product image plus textual instructions. This oversimplified data source fails to capture the complexity of real webpages (e.g., nested layout hierarchies, linked CSS/JavaScript files, cross-page resource dependencies). As a result, the webpage generation scheme proposed in this manuscript lacks relevance to practical production scenarios—generated HTML may only "superficially follow instructions" but cannot match the structural and functional integrity of real e-commerce webpages, undermining its practical value.

2. Web2Code’s benchmark (e.g., WCGB) evaluates generated webpages by comparing them with real webpages (e.g., measuring structural similarity, style consistency, and functional equivalence to real-world counterparts). However, MMWebGen’s evaluation solely relies on comparing generated webpages against textual instructions and the source image—it lacks a "real webpage baseline" for reference. This evaluation design cannot verify whether generated webpages meet the standards of practical web development (e.g., whether the layout is consistent with common e-commerce webpage conventions, or whether CSS styles are compatible with mainstream browsers).

3. The HTML code in MMWebGen-1k is generated via a "GPT-4o draft → Gemini-2.5-Flash refinement" pipeline, which fails to resolve inherent issues with synthetic data. Such as using GPT-4o as part of the LLM-as-a-judge for evaluation. This creates a potential conflict of interest—GPT-4o may implicitly favor HTML structures similar to those it generated, introducing subjective bias into the evaluation results and reducing the credibility of performance claims for the fine-tuned model.

**Questions:**

1. While the paper claims that "the benchmark and dataset will be publicly available", as of the review, no access links or release timelines are provided. Additionally, the code for model fine-tuning (e.g., BAGEL’s training configuration) and evaluation (e.g., LLM-as-a-judge implementation) is not disclosed. This lack of transparency makes it impossible for reviewers or other researchers to replicate the experiments, severely compromising the work’s scientific rigor.

2. The manuscript evaluates generated HTML code solely by checking "compliance with instructions" via LLMs, without any ground-truth HTML for comparison. This approach has two critical issues: (1) LLMs may have inherent biases (e.g., favoring certain layout styles or wording), leading to inconsistent ratings. (2) "Instruction compliance" does not equate to "practical usability"—generated HTML may follow instructions but still be non-renderable.

---

### Official Review · Reviewer_4UNo · 2025-10-31

**Soundness:** 3
**Presentation:** 3
**Contribution:** 3
**Rating:** 4
**Confidence:** 4

**Summary:**

MMWebGen proposes a benchmark for multimodal webpage generation in a product-showcase setting (130 prompts, 13 categories). It evaluates two workflows—editing-based and UM-based. Five metrics are used: WIF, WDQ, WCA (webpage) and VCIF, IPQ (image). The authors also release MMWebGen-1k and show BAGEL improves markedly after SFT.

**Strengths:**

1. Joint HTML+image generation with concrete instructions and source image mirrors real product pages.
2. The are five dimensions cover instruction following, layout, and visual consistency, and the paper provides evidence of correlation with human judgements, supporting their practical usefulness.
3. MMWebGen-1k and the fine-tuning experiments demonstrate improvements and a path for others to reproduce and extend the work.

**Weaknesses:**

1. Typos: "VCIT" in Line 216,221; "Claude sonnect 4" in Line 352,371.
2. Judge: GPT-4o and Gemini-2.5-Flash are used as LLM-as-a-judge while related systems are evaluated, inviting self-evaluation or style bias.
3. The paper labels two pipelines as “editing-based” and "UM-based", but in practice the UM-based pipeline is still a two-stage process—first generating the full HTML, then producing each image. This discrepancy with the abstract and intro’s claim of "co-generation".
4. The paper does not disclose randomness and decoding settings, though generation quality depends heavily on these hyper parameters.
5. The choice of the hyper parameter λ = 4 in the joint training objective in Section 4, which lacks a sensitivity ablation.

**Questions:**

1. Please fix the typos, and standardize metric and model names shown in Weaknesses.
2. How do you mitigate potential self-evaluation or style bias when GPT-4o and Gemini-2.5-Flash act as judges while related systems are being evaluated? Decouple judging by using a third-party judge or a multi-judge ensemble with majority vote.
3. What are the decoding hyperparameters (e.g., temperature, top-p, sampling steps) and random seeds used across runs?
4. Why was λ = 4 selected and how sensitive are results to this choice? Add a sensitivity ablation over λ and discuss the trade-off mentioned in the paper.

---

### Official Review · Reviewer_Vev3 · 2025-10-31

**Soundness:** 3
**Presentation:** 2
**Contribution:** 2
**Rating:** 4
**Confidence:** 4

**Summary:**

This paper proposes MMWebGen, a new benchmark designed to evaluate multimodal webpage generation capabilities. Focusing on a product showcase scenario, the authors compare editing-based and unified model-based approaches, showing respective strengths in webpage quality and visual consistency. The benchmark defines five evaluation metrics using the LLM-as-a-judge framework and validates them through human assessment. Furthermore, they construct the MMWebGen-1k dataset and fine-tune the open-source unified model BAGEL, achieving performance improvements.

**Strengths:**

- The paper addresses the underexplored task of multimodal webpage generation in complex, production-level scenarios. Its originality lies in jointly generating both HTML code and visual content, going beyond typical benchmarks that focus only on layout generation or textual webpage understanding.

- The paper provides a thorough comparative analysis of editing-based and UM-based approaches. It highlights their distinct advantages in webpage quality versus visual consistency, offering valuable insights into the trade-offs between modular and unified generation pipelines.

- By constructing the new MMWebGen-1k dataset and fine-tuning the open-source model BAGEL, the paper contributes to narrowing the performance gap between open- and closed-source models.

- The use of LLM-as-a-judge enhances the objectivity of evaluation, and the inclusion of human assessments further validates the clarity and reliability of the proposed metrics.

**Weaknesses:**

- Although MMWebGen aims to evaluate multimodal generation by combining text and image generation, the representativeness of its design remains unclear. The benchmark focuses on a narrow product showcase scenario with 13 predefined categories, which may limit generalization beyond this marketing-oriented context.

- In practice, most webpages rely on real product images rather than AI-generated ones, suggesting that the benchmark may not fully reflect real-world webpage creation needs. Furthermore, while the five proposed metrics are effective for assessing instruction following and visual quality, it seems they overlook aspects like functional correctness and usability, raising questions about whether the benchmark convincingly supports its claimed “production-level” motivation.

- While MMWebGen addresses a complex multimodal webpage generation task, the benchmark size (130 samples) is relatively small compared to prior benchmarks. This limited scale may constrain the statistical reliability of the evaluation and the generalization of its findings. Do the authors believe that a benchmark with only 130 samples is sufficient to provide reliable and representative evaluation results?

- After SFT, BAGEL’s IPQ slightly decreased. The authors attribute this to the possible inclusion of low-quality images and the lack of quality-based filtering in the dataset. This raises concerns about the overall quality and reliability of the MMWebGen-1k dataset used for fine-tuning.

- There is an inconsistency in notation: although the metric is defined as VCIF (Visual Content Instruction Following) in Section 2.2, Tables 1, 2, 3, and 5 refer to it as VCIT.

**Questions:**

- The paper states that GPT-4o and Gemini-2.5-Flash are used as evaluation models. Could this introduce evaluation bias, favoring outputs generated by the same model family?
- Among the three open-source UMs evaluated (BAGEL, Ovis-U1, and OmniGen2), only BAGEL was fine-tuned on the MMWebGen-1k dataset. Would the same fine-tuning strategy applied to Ovis-U1 or OmniGen2 lead to similar performance improvements?
- The paper presents quantitative improvements after fine-tuning BAGEL (Table 5), but no qualitative examples are shown. Could you provide visual comparisons to illustrate how fine-tuning improves webpage quality or visual consistency?

---

### Official Review · Reviewer_Xsj4 · 2025-10-31

**Soundness:** 2
**Presentation:** 2
**Contribution:** 2
**Rating:** 4
**Confidence:** 4

**Summary:**

This paper introduces MMWebGen, a novel benchmark for evaluating the ability of multimodal models to generate product webpages from source images and complex instructions. The work's primary contributions are the benchmark itself, the creation of the MMWebGen-1k fine-tuning dataset, and an in-depth analysis of the current capabilities and limitations of state-of-the-art models on this practical and challenging task.

**Strengths:**

The paper conducts the evaluation of multimodal models on a  more practical application—webpage generation—moving beyond simpler academic benchmarks.

The open-sourced MMWebGen-1k dataset could help community.

The paper is  clear and easy to read.

**Weaknesses:**

The paper primary offers a new dataset, but its modest scale raises questions about the benchmark's robustness. There is no methodological novelty; the work simply fine-tunes an existing model that still fails to match the performance of proprietary systems. This fine-tuning on a narrow domain also introduces a high risk of overfitting and limited generalizability.
Consequently, the paper occupies an uncertain position: it is not a landmark benchmark due to its limited scale, nor is it a compelling modeling paper due to its lack of novelty or state-of-the-art results.

**Questions:**

Regarding Figure 1, could the authors please clarify its intended purpose and the main takeaway for the reader?

---

### Note · Authors · 2025-11-14

**Comment:**

Thank you very much for the time and effort the reviewers and area chair devoted to evaluating our submission. We sincerely appreciate the constructive comments and valuable suggestions provided. They have given us important insights into how we can further improve the clarity, rigor, and contribution of our work. We have decided to withdraw the paper at this time in order to address these points and strengthen the manuscript. We are genuinely grateful for the reviewers’ feedback, which will guide our revisions moving forward.

**Withdrawal Confirmation:**

I have read and agree with the venue's withdrawal policy on behalf of myself and my co-authors.